# Green Wastes Mediated Zinc Oxide Nanoparticles: Synthesis, Characterization and Electrochemical Studies

**DOI:** 10.3390/ma13194241

**Published:** 2020-09-23

**Authors:** Enyioma C. Okpara, Omolola E. Fayemi, El-Sayed M. Sherif, Harri Junaedi, Eno E. Ebenso

**Affiliations:** 1Department of Chemistry, School of Physical and Chemical Sciences, Faculty of Natural and Agricultural Sciences, North-West University (Mafikeng Campus), Private Bag X2046, Mmabatho 2735, South Africa; ebrochima@gmail.com; 2Material Science Innovation and Modelling (MaSIM) Research Focus Area, Faculty of Natural and Agricultural Sciences, North-West University (Mafikeng Campus), Private Bag X2046, Mmabatho 2735, South Africa; Eno.Ebenso@nwu.ac.za; 3Center of Excellence for Research in Engineering Materials (CEREM), King Saud University, P.O. Box 800, Al-Riyadh 11421, Saudi Arabia; esherif@ksu.edu.sa; 4Electrochemistry and Corrosion Laboratory, Department of Physical Chemistry, National Research Centre, El-Buhouth St., Dokki, Cairo 12622, Egypt; 5Mechanical Engineering Department, College of Engineering, King Saud University, P.O. Box 800, Al-Riyadh 11421, Saudi Arabia; hjunaedi@ksu.edu.sa

**Keywords:** citrus peel extract, zinc oxide, nanoparticles, spectroscopy, screen plate carbon electrode, cyclic voltammetry

## Abstract

Zinc Oxide (ZnO) nanoparticles were prepared using a simple green synthesis approach in an alkaline medium, from three different extracts of citrus peels waste. The synthesized nano-crystalline materials were characterized by using ultraviolet-visible spectroscopy (UV-vis), x-ray powder diffraction (XRD), Fourier-transform infrared spectroscopy (FTIR), energy-dispersive x-ray spectroscopy (EDS), environmental scanning electron microscopy (ESEM), and transmission electron microscopy (TEM). UV-vis analysis of the nanoparticles showed broad peaks around 360 nm for the ZnO NPs (Zinc oxide nanoparticles) from three citrus peels’ extracts. ZnO NPs exhibited Zn–O band close to 553 cm^−1^, which further verified the formation of the ZnO NPs. A bandgap of 3.26 eV, 3.20 eV and 3.30 eV was calculated for the ZnO NPs from grape (ZnO NPs/GPE), lemon (ZnO NPs/LPE), and orange (ZnO NPs/OPE) peels extract, respectively. The average grain sizes of the ZnO nanoparticles were evaluated to be 30.28 nm, 21.98 nm, and 18.49 nm for grape (ZnO NPs/GPE), lemon (ZnO NPs/LPE), and orange (ZnO NPs/OPE) peel extract, respectively. The surface morphology and sizes of the nanoparticle were confirmed by ESEM and TEM analysis, respectively. Furthermore, the zeta potential of the as-prepared ZnO NPs from OPE, LPE, and GPE was −34.2 mV, −38.8 mV, and −42.9 mV, respectively, indicating the high stability of the nanoparticles. Cyclic voltammetric properties of the synthesized nanoparticles were investigated across extracts, and the results showed that the citrus peels extracts (CPE) mediated ZnO NPs modified screen plate carbon (SPC/ ZnO NPs/CPE) electrodes exhibited enhanced catalytic properties when compared with the bare SPCE. The electroactive areas computed from the enhancement of the bare SPCE was approximately three times for SPCE/ ZnO NPs/LPE, and SPCE/ZnO NPs/GPE, and two times for SPCE/ZnO NPs/OPE, higher than that of the bare SPCE. Comparison across the extracts suggested that the catalytic properties of the nanoparticles were unique in ZnO NPs from GPE.

## 1. Introduction

Nanotechnology involves methods of fabrication of materials at the lowest scale possible [1]. The primary purpose of nanotechnology is essentially nanoparticles synthesis [2]. Owing to their unique catalytic, mechanical, thermal, electrical and optical properties being different and better than their bulk counterpart, they have potential and growing applications in diverse fields [3,4]. The addition in surface area to volume ratio, as a rule, changes the catalytic, and thermal properties of the material [5].

The choice of electrode material is very pivotal in electrochemical sensing technology and has consequently been extensively studied [6]. The combined effect of nanomaterials and electrochemical sensor materials would result in advantageous and expanding sensing applications [7,8]. Nanostructured material sensors exploit the advantages of the amplified electrode (electroactive) surface area, fast rate of electron transfer, and enhanced rate of mass transport in comparison to bulk electrode materials [7]. Additionally, the use of nanomaterials presents new simple strategies to fabricate very cost-effective electrodes, with improved sensing parameters that use fewer materials and also generate less waste [9].

Zinc oxide nanoparticles have many attractive properties, such as large binding energy, wide bandgap, and high piezoelectric property [4]. They found applications in a huge number of areas such as optoelectronic devices, laser devices, electromagnetic coupled sensor, and surface acoustic wave devices [10,11,12,13,14,15]. They also have remarkable application in biomolecular detection diagnostics, and microelectronics [16,17]. As expected, nanoparticles of metal oxide, have numerous better and substantial possibilities of application, which include, among others, cell line studies, anti-microbial, and degradation of dye properties [2]. Interestingly, Zinc oxide nanoparticles (ZnO NPs) have a bandgap of 3.37 eV that is also significant for several applications [1,4]. In literature, ZnO nanoparticles are synthesized from conventional methods such as chemical reduction [17], laser ablation [18], solvothermal [19], inert gas condensation [20], sol-gel method [21], etc. These methods require some toxic chemicals, high pressure, laser radiation, and inert gases such as helium compared to green synthesis methods. Additionally, some of these conventional methods are expensive, not easy to operate, and require much attention during the process and a particular type of vessel-like polypropylene vessel for nanoparticles synthesis [4]. Hence, the formation and engineering of ZnO NPs would profit from the emergence of nontoxic, clean, cheap, reliable, and environmentally acceptable “green chemistry” route for nanoparticles synthesis [1]. The green route synthesis of ZnO NPs of various sizes, in the range of 1 and 70 nm, and morphologies investigated utilizing fungi, bacteria and plant extracts [22,23,24,25].

Legislation and concerns on environmental safety about the disposal of organic wastes control are rising [26]. Unwanted peels of fruit—such as peels from pomegranates, lemon, and oranges—shells of eggs, and peels of shrimp, in addition to the organic components of solid wastes from the municipality, could be used in the growing subject of nanotechnology [27,28,29]. For instance, in the process of production of orange juice, roughly 50–60% of the fruit waste processed is converted to wastes: seeds, peels, and membrane remains [30]. Thus, a chunk portion, out of large amounts of orange peels annually produced was often poorly managed, which puts forth severe waste consequences on the environment [31]. The recycling of citrus species peel wastes from the industry is critical to realize the dual intents of material recycling and waste control, hence providing profitable products while keeping the ecosystem from the detrimental consequences generated as a result of the accretion of that kind of waste. The citrus set is a large family of fruits [32]. Citrus fruits and peels have a significant amount of bioactive compounds such as carotenoids, coumarins, limonoids, alkaloid, tannins, saponins, phenol, amino acid, protein and flavonoids [1,33,34].

This study, therefore, describes thriving green mediated reduction of zinc salt to ZnO NPs, its capping and stabilization, and subsequent characterization of the as prepared nanoparticles from *Citrus x paradise* (acidic grapefruit), *Citrus sinensis* (sweet orange) and *Citrus limon* (Lemon) and evaluation of electrochemical properties that can aid electrochemical applications as sensors.

## 2. Materials and Methods

### 2.1. Materials

The precursor plant part used were peels of *Citrus x paradise* (acidic grapefruit), *Citrus sinensis* (sweet orange), *Citrus limon* (Lemon)). Chemically pure zinc acetate [(CH_3_COO)_2_ Zn·2H_2_O; ≥98% purity] and Potassium hexacyanoferrate (III), (K_3_[Fe(CN)_6_]; 99% purity) were purchased from Sigma-Aldrich, Chemie GimbH, Steinheim, Germany. Sodium hydroxide (NaOH; 99% purity) and chemically pure HCl (30%) were purchased from Emsure Iso, Merck KGaA, Darmstadt, Germany and Promark Chemicals (Robertsham, Johannesburg, South Africa), respectively. Potassium chloride was purchased from Metrohm Ltd., cH-9100 Hensau, Switzerland. Pure distilled water was obtained from the laboratory. and KCl were used to prepare 0.1 M solution of the probe buffer solution. 1 M NaOH and 1 M HCl were used to set the pH value of the buffer solution to 7.0.

### 2.2. Synthesis of ZnO NPs

The extraction of the citrus peels followed the procedure described in our previous work [35] and another literature report [36]. The filtrate was later kept in the refrigerator for subsequent use. Zinc acetate (200 mL of 0.01 M solution) was added into three separate conical flasks heated to a temperature above 60 °C. Typically, 20 mL of each of the citrus peels’ extract from orange, grape and lemon were introduced each separately into the three flasks, while maintaining the temperature conditions as shown in Figure 1. The colorless zinc acetate solution immediately turned pale yellow, largely influenced by the color of the extract. After 5 min of vigorous stirring of the mixture, 10 mL of 1 M aqueous solution of sodium hydroxide (NaOH) was also added to each of the CPE and zinc acetate mixtures. Immediate color change from pale yellow to dirty cream white confirmed the formation of the ZnO NPs. The mixtures were stirred for four hours while checking the UV-vis spectrum data to ascertain optimum reaction time. After 4 h, each mixture was cooled to room temperature, centrifuged at 6000 rpm for 20 min and washed twice with distilled water. The white precipitate was assembled and ovum dried. Subsequently, the resultant white powder was obtained and stored for further characterization.

The material characterization of the samples was performed with UV–Vis Uviline 9400 (Sl Analytics, Hattenbergstr.10, D-55122 Mainz, Germany) to study the optical properties of the ZnO NPs synthesized. FTIR (Opus Alpha-P, Brucker corporation, Billerica, MA, USA) was employed to evaluate the functional groups that interacted with the precursor acting as a reducing or capping agent or both. The powdered samples were characterized in the scale of 4000–400 cm^−1^. X-ray diffraction (XRD) angles were also investigated with the aid of Röntgen PW3040/60 X’Pert Pro X-ray diffractometer (Germany) having Ni-filtered Cu Kα radiation (λ = 1.5405 Å) at a rate of scanning of 2° min^−1^ and 2θ ranging from 10° to 90°, to examine the crystallinity and average particle size (using Debye–Scherrer equation) of the ZnO NPs. The morphology of the surface of the synthesized ZnO NPs/CPE was characterized with Quanta FEG 250 Environmental Scanning electron microscope (ESEM, ThermoFisher Scientific, OR, USA) using an acceleration voltage of 15.0 kV while simultaneously detailing the energy dispersive spectrum (EDS). Transform Electron Microscopy (TEM) with JEOL2100 instrument (Joel Ltd., Peabody, MA, USA) fitted with a LaB 6 electron gun and images taken with a Gatan Ultrascan digital camera was employed to investigate the morphology of the surface of ZnO NPs. DropSense (analyzed with Dropview 200 software, Metrohm South Africa (Pty) Ltd., Johannesburg, South Africa) fitted with SPCE of inner carbon working electrode (WE) diameter of 4 mm, an Ag pseudo-reference (Ag/AgCl) electrode (RE) and carbon counter electrode (CE), with a potential window of −1.0 to 1.2 V, and redox currents of −200.00 to 200.00 µA was used for the CV electrochemical studies.

### 2.3. Fabrication of the SPCE

Each of the screen plate carbon (disposable) electrodes were modified by drop dry approach. Then, 2 mg of ZnO NPs were dissolved in dimethylformamide and sonicated at room temperature. Typically, the resultant was dropped to cover the DropSense SPC working electrode center (Metrohm South Africa (Pty) Ltd., Johannesburg, South Africa) of 4 mm diameter and air-dried.

### 2.4. Voltammetric Measurements

The electrochemical properties of the synthesized nanoparticles were hence evaluated by using cyclic voltammetry (CV) in 10 mM K_3_[Fe(CN)_6_] probe prepared in 0.1M KCl at pH ≈ 7.0. The solution of the probe is put on the electrodes to cover it enough and scanned using the Dropview 200 software (Metrohm South Africa (Pty) Ltd., Johannesburg, South Africa). All experiments were performed at room temperature, with the potential step of 0.01 V. The measurements were simple and straightforward.

## 3. Results and Discussion

### 3.1. Characterization of the As-Prepared ZnO NPs

#### 3.1.1. UV-Vis

The reduction in pure Zn^2+^ ions was confirmed by measuring the UV-vis spectrum while the reaction was taking place. The spectrum was scaled on a wavelength from 300 to 800 nm of the UV-Vis absorption spectra, as shown in Figure 2, to evaluate the optical properties of the nanoparticles. The synthesized ZnO NPs/GPE showed a broad absorbance peak with redshifts at 348 nm, 360 nm, and 366 nm within 10 min, 1–3 h, and 4–5 h (Figure 2a) of synthesis, which is ascribed to the agglomeration of ZnO NPs, leading to the rise in the size of the particles with more time of interaction between the plant and metal salt precursors [37]. Each time there is redshift, the absorbance drops and continued to increase until another redshift. LPE/ZnO NPs showed a consistent increase in absorbance at a peak around 360 nm (Figure 2b), with little red and bled shifts. The LPE ZnO NPs characteristic peak was observed within the first 10 min, and the concentration of the ZnO NPs continued to increase until the 4 h (Figure 2b). After 4 h, the absorbance dropped with a slight redshift. The OPE/ZnO NPs (Figure 2c) was formed also within the first 10 min, showing a similar shift as the GPE/ZnO NPs (Figure 2a). They showed a red peak shift from 346 nm after 10 min to 360 nm after 5 h. At the optimum time, there was a slight blue shift in the absorbance peak from 360 nm to 359 nm (*Eg* = 3.54 eV), indicating reduced ZnO NPs size and quantum confinement effects [38]. All the absorption peaks were consistent with that of previous work from the literature [39,40]. Figure 2a–c also represents a comparison of the CPE/ZnO NPs at their respective optimum times. At these optimum times, they all seem to have the same absorption peak of about 360 nm with very little variations. However, the concentration of the CPE/ZnO NPs based on absorbance is in this order GPE/ZnO NPs < OPE/ZnO NPs < LPE/ZnO NPs. ZnO as reported in the literature to have a broad direct bandgap of 3.37 eV and a semiconductor. The energy bandgap (*Eg*) between the conduction and the valence band are represented by Equations (1) and (2):(1)$${\left(\alpha hv\right)}^{2}=A\left(hv-Eg\right)$$
where hν represents the energy of the photon, and *α* represents the absorption edge [41].
(2)$$\alpha =\frac{1}{d}In\left(\frac{1}{T}\right)$$

From the plots of (αhν)^2^ versus hυ (E, eV) of the CPE/ZnO NPs using the UV spectra, (graph not shown), the *Eg* values were determined and discovered to be 3.26, 3.20 and 3.30 eV for GPE/ZnO NPs, LPE/ZnO NPs, and OPE/ZnO NPs, respectively. These values are very comparable to those presented in earlier literature [36]; however, less than 3.37 eV which could be attributed to changes in lattice parameters [38,42]. The computed *Eg* result reveals that the energy bandgap for all CPE/ZnO NPs is roughly the same, but specifically closely follows this order: LPE/ZnO NPs < GPE/ZnO NPs < OPE/ZnO NPs. The changes in the coefficient of absorption, α in relation to photon energy are reported in Figure 3a and signifies that the absorption coefficient rises with the increasing energy of the photon and falls exponentially with higher wavelengths, as shown in Figure 3b. This pattern is symbolic of many semiconductors and may be ascribed to several reasons spiraling from internal electric fields within the crystal, defectiveness leading to a twist that results in the deformation of lattice, to fixed scattering of carriers of charges by the phonons [43,44,45]. The highest absorption coefficient is sighted in the UV region within the range of 10^−7^–10^−6^ (nm)^−1^ for all the green waste initiated NPs, with LPE/ZnO NPs, (Figure 3a) having the highest absorption coefficient, seconded by OPE/ZnO NPs, and then GPE/ZnO NPs. α rose exponentially in all of the CPE/ZnO NPs with rising photon energy up to about 3.5 eV, dropped briefly and subsequently continued increasing exponentially.

#### 3.1.2. FTIR

FTIR spectroscopy was used to investigate the various phytochemicals that could be responsible for the bioreduction of metal precursor into corresponding metal oxide NPs and their following growth inactivation under the capping effect of the biomolecules. The FTIR spectrum, displayed in Figure 4, represents the powdered sample of citrus peel extract initiated ZnO NPs. The peak at 3378 cm^−1^ corresponds to the strong broadband of –OH group commonly found in organic molecules attached to the surface of the NPs. The other peaks at 1560 cm^−1^, 1398 cm^−1^, 1040 cm^−1^ and 870 cm^−1^ correspond to carbonyl stretches, C–OH bending (in-plane), vibrations of carboxylic acids and bending frequencies of C–O and –CH out of plane bending vibration of trans or E-alkene, respectively [46,47,48]. The ZnO NPs spectrum exhibited a typical Zn–O stretching at ~553 cm^−1^ for all the synthesized CPE/ZnO NPs, which verifies the formation of ZnO NPs [49].

#### 3.1.3. Zeta Potential

Zeta potential (ζ) is an essential device for defining the charges on the surface of NPs in aqueous form and envisaging the long term stability [50]. The ζ of the ZnO NPs was also evaluated with a Malvern instrument to ascertain the stability of the NPs as represented in Figure 5 below. The long term stability of all the NPs was found to be high from instrumental study using the Malvern multipurpose titrator (Figure 5a–c). The ζ values were −34.2 mV, −38.8 mV, and −42.9 mV for OPE/ZnO NPs, LPE/ZnO NPs, and GPE/ZnO NPs, respectively. All were very stable ZnO NPs with zeta values for GPE/ZnO NPs > LPE/ZnO NPs, and OPE/ZnO NPs. The graph in Figure 5d shows the relationship between the zeta potentials and the particles size. The higher particle-sized NPs showed higher zeta potential value and excellent stability.

#### 3.1.4. SEM

The SEM image characterizing the morphology of the as-prepared ZnO NPs is presented in Figure 6. It shows that in general they are agglomerated spherical clusters with smooth surfaces, apparently void of cracks. However, In Figure 6c, ZnO NPs/OPE shows more closely parked agglomeration while in Figure 6b, LPE/ZnO NPs has a mixture of few (about two or three) rod-like structures. This image agrees with previous work on the biological synthesis of ZnO NPs [48].

The elemental composition of the CPE/ZnO NPs from the three extracts was considered and summarized in Table 1. The result shows that LPE/ZnO NPs has highest percentage composition of Zn, followed by OPE/ZnO NPs and then GPE/ZnO NPs. However, the difference in composition of Zn between the highest and lowest is significantly low (6.03%).

#### 3.1.5. TEM

TEM analysis was carried out on the CPE/ZnO NPs powders to estimate the particle sizes. The results confirmed the agglomerated tiny irregular shaped nano-sized particles with LPE/ZnO NPs tending to be more clustered than the two other extracts mediated ZnO NPs (Figure 7). Image J bundled with 64-bit Java 1.8.0_112 software was used to analyze the particle size distribution of the as prepared NPs. GPE/ZnO NPs, LPE/ZnO NPs, and OPE/ZnO NPs give an average particle size of 22.00 ± 4.78 nm, 26.90 ± 3.76 nm, and 19.63 ± 2.89 nm, respectively, and represented in Figure 8. The size distribution ranges from 10.89–54.41 nm, 10–40 nm, and 11.31–31.68 nm for GPE/ZnO NPs, LPE/ZnO NPs and OPE/ZnO NPs, respectively. These values vary slightly from the average particle size of CPE/ZnO NPs powder gotten from XRD analysis.

#### 3.1.6. XRD

The powder X-ray diffraction (PXRD) patterns of the as-prepared ZnO NPs were measured at room temperature, using a powered Röntgen PW3040/60 X’Pert Pro X-ray diffractometer having Ni-filtered Cu Kα radiation (λ = 1.5405 Å) at a scan rate of 2° min^−1^ and 2θ ranging from 10° to 90°. The XRD pattern for the three samples is presented in Figure 9.

Figure 9, demonstrates the X-ray diffraction (XRD) patterns, signifying that the ZnO presents a hexagonal wurtzite structure. The broadening of the peak in the XRD pattern is an indication that nanocrystals of small dimension exist in the samples. The strong diffraction peaks further show the appropriate crystalline nature of the synthesized nano-scaled-particles—subsequently to the crystallization of ZnO in the wurtzite pattern, where the O-atoms were positioned hexagonally and packed tight to the zinc atoms taking up half of the tetrahedral spots. Zn and O-atoms coordinate in a tetrahedral way to each other and, consequently, have a parallel position. Hence, Zn structure is open having all the octahedral and half of the tetrahedral sites left empty [51]. Across the different extracts, the ZnO NPs from OPE and LPE exhibited patterns with well-defined index peaks comparably marched with the bulk wurtzite hexagonal well-crystalline ZnO (Table 2), than that of the ZnO NPs/GPE.

The lattice constants, a = b = 3.2656 Å and c = 5.2095 Å, c/a = 1.5997, and diffraction peaks corresponding to the planes (100), (002) (101), (102), (110), (103) obtained from X-ray diffraction data are consistent with the JCP2 (card number 36-1451 for ZnO). The interplanar spacing (d_hkl_) calculated from XRD is compared with JCP2 data card and equivalent (hkl) planes, percentage of variation of d_hkl_ for some major XRD peaks are summarily presented in Table 2. According to Bragg’s law Equation (3):(3)nθ=2dsinӨ
with n as the diffraction order (often n is equal to 1), λ represents the X-ray wavelength (λ = 0.15405 nm), and d is the spacing between planes of specific Miller indices h, k and l. In the ZnO hexagonal pattern, the subsequent relation exists between the spacing the plane (d), the Miller indices and the lattice constants a, c represented by Equations (4)–(7):(4)$$\frac{1}{d_{\left(\mathrm{hkl}\right)}^{2}}=\frac{4}{3}(\frac{h^{2}+\mathrm{hk}+k^{2}}{a^{2}})+\frac{l^{2}}{c^{2}}$$

Using first-order approximation of n = 1;
(5)$${\mathrm{Sin}}^{2}\theta \;=\frac{{\;\lambda }^{2}}{4a^{2}}\left[\left(\frac{4}{3}\left(h^{2}+k^{2}+\mathrm{hk}\right)\right)+\left(\frac{a^{2}}{c^{2}}\right)l^{2}\right]$$
lattice constant (a) of (100) plane may be computed from the relation [52]:(6)$$a=\frac{\lambda }{\sqrt{3}\sin\;Ө\;)\;}$$

For the plane (002), the lattice constant c may be computed using [52]
(7)$$c=\frac{\lambda }{\sin Ө\;}$$

The lattice constants computed, a = b = 3.2656 Å and c = 5.2095 Å, c/a = 1.5997) and diffraction peaks corresponding to the planes (100), (002) (101), (102), (110), (103) gotten from X-ray diffraction data are consistent with the JCP2 (card number 36-1451 for ZnO). The interplanar spacing (d_hkl_) calculated from XRD is compared with JCP2 data card and equivalent (hkl) planes; the percentage of variation of d for some major XRD peaks is summarily presented in Table 2. The calculated values of d_hkl_ are also comparable with a previous work [47]—this present work exhibiting lower contraction with the JCPDS data. The average size of crystallite was estimated from XRD peak width of the preferential plane (101) using the Debye–Scherrer [53] equation:(8)$$D=\frac{K\lambda }{\left({ß}_{\mathrm{hkl}}\right)\cos\theta }$$
where ß_hkl_, λ, K and Ө are the full width at half maximum (FWHM), 0.1540, 0.9 and Bragg’s angle, respectively. The calculated mean size for the engineered zinc oxide NPs was 30.28 nm, 21.98 nm and 18.49 nm for grape, lemon and orange peel extract mediated ZnO NPs respectively.

The XRD analysis resulted in GPE/ZnO NPs having the maximum size, while the LPE/ZnO NPs have the maximum size distribution using TEM analysis. The disparity in the maximum diameters found for using XRD and TEM analysis could be attributed to possibly (1) inherent limitation associated with using XRD, as it accounts for just average particle size diameter that caused diffraction, (2) the larger size distribution of GPE/ZnO NPs over LPE/ZnO NPs, that could lower mean average and (3), TEM analysis takes into consideration of an particular area of photo and the use of image J software analyzer that also presupposes a completely spherical shape [38]. However, the images are not absolutely spherical. Hence, such discrepancies in the values of the maximum diameter are possible. The density (δ) of dislocation, which is equivalent to the amount of defects in the sample represents the length of dislocation lines per unit volume of the crystal and is computed from the relation [53].
(9)$$\delta =\frac{1}{D^{2}}$$
and D represents the crystallite size. The dislocation density for ZnO NPs/GPE, ZnO NPs/LPE and ZnO NPs/ OPE calculated are 11 × 10^−4^, 20.7 × 10^−4^, and 29 × 10^−4^ (nm^−2^), respectively.

The Zinc to oxygen (Zn-O) bond length L is obtained from Equation (10) [54]:(10)$$L=\sqrt{\left(\frac{a^{2}}{3}+{\left(\frac{1}{2}-u\right)}^{2}c^{2}\right)},$$
where u represents the positional parameter in the wurtzite structure and is a measure of the amount by which each atom is displaced with respect to the next along the ‘c’ axis [54]. ‘u’ is calculated from the relation:(11)$$u=\frac{a^{2}}{3c^{2}}+0.25$$

The relatively lower c/a ratio results in increased value of ‘u’, which consequently indicates that the four tetrahedral distances will remain nearly constant through a possible distortion in the tetrahedral angles [51]. The Zn–O length computed for the ZnO NPs/CPE was 1.9847 Å, while the Zn–O reported in the literature is 1.9654 Å [51]. Another literature report described Zn–O bond length in the unit cell of ZnO and adjoining atoms to be 1.9767 Å [55]. Hence, the estimated length of the bond (L) is consistent with the Zn–O bond length in the unit cell with 0.4% difference. Hence, this minor variation could be attributed to a possible distortion in the tetrahedral angles.

### 3.2. Electrochemical Characterization of the Citrus Peels Extracts

#### 3.2.1. Cyclic Voltammetry

The electrochemical properties of the synthesized nanoparticles were studied using cyclic voltammetry (CV) in a 10 mM probe in 0.1 M KCl at pH ≈ 7. The SPCE electrodes used comprise of carbon counter electrode, an Ag pseudo-reference (Ag/AgCl) electrode, and four working electrodes. The working electrodes include, bare SPCE, ZnO NPs/GPE, ZnO NPs/LPE, and ZnO NPs/GPE modified electrodes to get SPCE/GPE/ZnO NPs, SPCE/LPE/ZnO NPs, and SPCE/GPE/ZnO NPs, respectively. The voltammogram of the bare screen plate carbon electrode (SPCE) and modified counterparts of SPCE/GPE/ZnO NPs, SPCE/LPE/ZnO NPs, and SPCE/OPE/ZnO NPs, over a potential window of −1.0 to 1.2 at a scan rate of 100 mV/s was recorded. A common potential step of 10 mV was used in all the CV. The comparative voltammograms shown in Figure 10 showed a well-defined redox peak at the bare SPE, but with a lower current response in comparison to the modified SPCE/ GPE/ZnO NPs, SPCE/LPE/ZnO NPs and SPCE/OPE/ZnO NPs in (Table 3). There is a considerable enhancement in the peak anodic current for all the SPCE/CPE/ZnO NPs in this order: SPCE/GPE/ZnO NPs < SPCE/LPE/ZnO NPs < SPE/OPE/ZnO NPs. The peak anodic currents for the SPCE/ZnO NPs/GPE, SPE/ZnO NPs/LPE and SPCE/ZnO NPs/OPE are approximately 1.3, 1.4 and 2.1 times greater than the bare SPE. The separation of the anodic and cathodic peak potential (∆Ep)—also summarized in Table 3—are higher than the value of 0.0598 V/n (n = 1), suggesting that the electrochemical behavior of probe under optimum situations is a quasi-reversible one-electron process [56,57]. The formal potential, indicating the spontaneity [58] and activity coefficient [59] of the electrodes in the redox reaction, was calculated to be 0.085 V, 0.145 V, 0.185 V and 0.195 V for the bare SPE, and modified SPE/ GPE /ZnO NPs, SPE/LPE/ZnO NPs and SPE/OPE/ZnO NPs, respectively, following the same enhancement order. The peak current ratio is more than unity for the SPE/GPE/ZnO NPs, SPE/OPE/ZnO NPs, and SPE/ZnO NPs LPE, which resembles a quasi-reversible system [56]. However, the ratio of anodic to cathodic peak currents for SPE/ZnO NPs/LPE was closer to unity than the rest of the as-prepared ZnO NPs, and resembled an ideal reversible system.

#### 3.2.2. Effect of Scan Rate

A number of voltammograms of 10 mM K_3_[Fe(CN)_6_] prepared in 0.1 M KCl at pH ≈ 7 and scan rate (10–200 mV/s) were recorded and represented in Figure 11. It was noticed that the anodic and cathodic peak currents increased linearly with the increase in the scan rate. There was a positive shift in the anodic peak potential for all the modified SPC electrodes, while on the cathodic peak potential, a shift towards the negative potential, with increasing scan rate was observed. This observation strongly indicates a redox process that is diffusion controlled.

A linear pattern was observed in a plot of peak currents I_pa_/I_pc_versus ν^1/2^ (Figure 12, Figure 13 and Figure 14) represented by the equations:(12)$$I_{pa}=507.04\;\nu^{\frac{1}{2}}-34.90;R^{2}=1.0;I_{pc}=-407.39\nu^{\frac{1}{2}}+22.96;R^{2}=1.0\;(\mathrm{SPCE}/\mathrm{ZnONPs}/\mathrm{LPE}),$$
(13)$$I_{pa}=544.40\nu^{\frac{1}{2\;}}-57.84;R^{2}=0.99;I_{pc}=-423.343\nu^{\frac{1}{2}}-33.55;R^{2}=0.99\;(\mathrm{SPCE}/\mathrm{ZnO}\;\mathrm{NPs}/\mathrm{GPE}),$$
(14)$$I_{pa}=429.83\nu^{\frac{1}{2\;}}-38.30;R^{2}=0.99;I_{pc}=-306.04\nu^{\frac{1}{2}}-5.64;R^{2}=0.99\;(\mathrm{SPCE}/\mathrm{ZnO}\;\mathrm{NPs}/\mathrm{OPE}),$$

The observed linear plot over the range of the study indicates that the process is still diffusion controlled. Moreover, a linear relationship was gotten from the plot of I_p_ vs. ν for the redox peaks (Figure 12d–f) [59,60]. Furthermore, a plot of log (Ip) vs. log ν, was examined (Figure 13d–f). Linear relationships were also observed as represented by the linear equations and resulting in slopes 0.7, 0.91, and 0.96 for SPCE/GPE/ZnO NPs, SPCE/LPE/ZnO NPs, and SPCE/OPE/ZnO NPs, respectively, which are greater than the theoretical values of 0.5 for a diffusion-controlled process [61].
(15)$$\log I_{p}\;=0.70\log\;\nu \;+0.69;R^{2}=1\;\mathrm{SPCE}/\mathrm{ZnO}\;\mathrm{NPs}/\mathrm{GPE}$$
(16)$$\log I_{p}\;=0.91\log\;\nu \;+0.22;R^{2}=0.99\;\mathrm{SPCE}/\mathrm{ZnO}\;\mathrm{NPs}/\mathrm{LPE},$$
(17)$$\log\;I_{p}\;=0.96\log\;\nu \;+0.06;R^{2}=0.9\;\mathrm{SPCE}/\mathrm{ZnO}\;\mathrm{NPs}/\mathrm{OPE}$$

The active surface area of the SPCE/CPE/ZnO NPs modified electrodes were evaluated using the Randles–Sevcik Equation (14) [62].
(18)$$\mathrm{Ip}=(2.69\times 10^{5)}{\mathrm{An}}^{\frac{3}{2\;}\;}D_{R}^{\frac{1}{2}}{C\upsilon }^{\frac{1}{2}}$$

n = number of electrons transferred,

A = active surface area (cm^2^),

D_R_ = Coefficient of diffusion, (cm^2^/s),

C = concentration (mol/cm^3^), for probe used, (n is equal to 1),

D = 7.6 × 10^−5^ cm^2^ s ^−1^,

υ = scan rate (V/s).

From Equation (18), the active surface area (A) is proportional to the value of Ip. At the scan rate of 100 mV/s and the corresponding peak currents as shown in Table 3, using the 10mM (converted to mol/cm^3^) concentration of the probe, the value of A was calculated from the Randle’s slope for the SPE, SPE/GPE/ZnO NPs, SPE/LPE/ZnO NPs and SPE/OPE/ZnO NPs and found to be 0.82, 2.38, 2.20 and 1.86 mm^2^, respectively.

#### 3.2.3. Rate of Transfer

The investigation of rate of transfer of electron in the electrolyte at the electrode–solution interface is an essential matter in electrochemistry. According to Laviron’s theory [63], the apparent charge transfers rate constant, ks, and the charge transfer coefficient, α, of a surface confined redox couple could be measured from the cyclic voltammograms using the congruent changes in the anodic and cathodic peak potentials as a function of the logarithm of scan rate. From the equation of the plot of peak potentials (Ep) vs. log ν (Figure 14a–c), the electron transfer coefficients (α) can be computed using the equations, over the linear range [64,65]:(19)$$E_{\mathrm{pa}}=a+\frac{2.303\mathrm{RT}}{\left(1-\alpha \right)n_{\alpha }F}\log\nu ,$$
(20)$$E_{\mathrm{pc}}=b-\frac{2.303\mathrm{RT}}{{\alpha n}_{\alpha }F}\log\nu ,$$
where E_pa_ is anodic peak potential and E_pc_ is cathodic peak potential, respectively. The calculated values of α for SPE/ZnO NPs/GPE, SPE/ZnO NPs/LPE, and SPE/ZnO NPs/OPE, respectively, are 0.60, 0.57 and 0.60. The values show that the transfer electron coefficient for SPE/GPE/ZnONPs = SPE/OPE/ZnONPs > SPE/LPE/ZnO NPs.

The heterogeneous rate of electron transfer in the redox probe may be calculated using the Equation (21) [65]:(21)$${\log\;k}_{s}=\alpha \log\;\left(1-\alpha \right)\;+\left(1-\alpha \right)\log\;\alpha \;-\log\frac{\mathrm{RT}}{\mathrm{nF}\nu }-\alpha \left(1-\alpha \right)\frac{\mathrm{nF}∆E_{p}}{\mathrm{RT}}$$
where n = number of electrons transferred; F = Faraday’s constant (96,500 C); T = Temperature (K)

R = Gas constant (8.3145 J. mol^−1^. K^−1^)

The rate constant for the electron transfer process is dependent on the electrode material nature. Given $n_{\alpha }$ and the corresponding values of α, from Equation (19) and (20), temp of 25 °C, the values of ks for the SPCE/GPE/ZnO NPs, SPCE/LPE/ZnO NPs and SPCE/OPE/ZnO NPs were calculated to be 0.0059 s^−1^, 0.075 s^−1^ and 0.0021 s^−1^, respectively. The rate of electron transfers as computed follows that ks for SPCE/ZnO NPs/LPE > SPCE/ZnO/GPE > SPE/ZnO NPs/OPE. The value of ks for SPE/GPE/ZnO NPs and SPE/LPE/ZnO NPs show that the transfer of electron is fast and the process is reversible, while the value of ks for SPE/OPE/ZnO NPs suggests that the electron transfer is slow and the process is a more quasireversible one [66].

The average surface concentration (Γ) of the probe on the surface of the SPE/CPE/ZnO NPs may be calculated from the slope of (I_p_) vs. (ν) (Figure 12d–f) by using Equation (22) [66,67].
(22)$${\;I}_{p}=\frac{n^{2}F^{2}A\Gamma \nu }{4\mathrm{RT}}$$
where A is the electrode surface area in m^2^, I_p_ is peak current in amperes, and every other parameter remains the same.

The relations between the peaks current is represented by the following equation:(23)$$I_{\mathrm{pa}}=932.57\nu +21.87;R^{2}=1.0\;(\mathrm{SPE}/\mathrm{LPE}/\mathrm{ZnO}\;\mathrm{NPs}),$$
(24)$$I_{\mathrm{pa}}=962.47\nu +11.64;R^{2}=0.99\;(\mathrm{SPE}/\mathrm{GPE}/\mathrm{ZnO}\;\mathrm{NPs}),$$
(25)$$I_{\mathrm{pa}}=745.75\nu +15.54;R^{2}=0.99\;(\mathrm{SPE}/\mathrm{OPE}/\mathrm{ZnO}\;\mathrm{NPs})$$

Given electrode diameter of 4 mm, with A equal to 1.2566 × 10^5^ m^−2^, the values of Γ calculated from the slope of the plot for the three SPE/ZnO NPs/CPE were 2.037 nmol cm^−2^, 1.973 nmol cm^−2^, and 1.578 nmol cm^−2^ for SPCE/GPE/ZnONPs, SPCE/LPE/ZnONPs, and SPCE/OPE/ZnONPs, respectively.

The peak potential, E_p_, showed a linear range for log ν, as shown presented in Figure 14a–c. The slopes of E_p_ vs. log ν for the modified electrodes of SPCE/GPE/ZnO NPs, SPCE/LPE/ZnO NPs and SPCE/OPE/ZnO NPs are 0.15, 0.17 and 0.18 Vdec^−1^, respectively.
(26)$$E_{p}=\frac{b}{2}\log\;\nu \;+\mathrm{constant}$$

The Tafel slopes (b) could be calculated according to the Equation (26) for totally irreversible diffusion-controlled process [67,68]. Hence, the values of b for SPE/ZnONPs/GPE, SPE/ZnONPs/LPE and SPE/ZnONPs/OPE electrodes are 300, 340 and 360 mVdec^−1^, respectively.
(27)$$E_{\mathrm{pa}}=0.17\log\left(\nu \right)+0.02;R^{2}=0.98\;(\mathrm{SPE}/\mathrm{ZnO}\;\mathrm{NPs}/\mathrm{LPE})$$
(28)$$E_{\mathrm{pa}}=0.15\log\left(\nu \right)-0.08;R^{2}=0.98\;(\mathrm{SPE}/\mathrm{ZnO}\;\mathrm{NPs}/\mathrm{GPE})$$
(29)$$E_{\mathrm{pa}}=0.18\log\left(\nu \right)-0.06;R^{2}=0.99\;(\mathrm{SPE}/\mathrm{ZnO}\;\mathrm{NPs}/\mathrm{GPE})$$

These Tafel values are, however, greater compared to the theoretical value of 118 mVdec^−1^ for a one-electron process engaged in the rate-determining step. This implies the existence of either adsorption or the interference of reaction intermediates on the surface of the electrode and is less obvious in the case of SPCE/LPE/ZnO NPs [68].

#### 3.2.4. Stability Studies

Several scans were run at a particular scan rate of 25 mV/s, as presented in Figure 15, to investigate the stability of the modified SPE towards the oxidation of the ferricyanide probe. It was discovered that the redox peak current (I_pa_/I_pa_) continued to increase gradually while the peak voltage continues to shift gradually towards the positive values at the anode and negative values at the cathode. After about 10 sans, SPCE/GPE/ZnO NPs, SPCE/LPE/ZnO NPs and SPE/OPE/ZnO NPs were found to have all experienced anodic peak increment of about 50%, 66.7% and 58.3%, respectively. This can be attributed to increasing electroactive surface area of the modified electrode following more interaction of the probe with time, despite constant scan rate. The voltammograms of the modified SPE in 0.1 M KCl, 10 mM K_3_[Fe(CN)_6_] solution at pH ≈ 7 over a potential window of −1 to 1.2 V, nevertheless show that the SPE/CPE/ZnO NPs have good stability. The redox peaks marked Io/Ir represent the redox couple of Zn^2+^/ZnO with anodic and cathodic peak potentials (0.93 V, 0.78 V), (0.91 V, 0.78 V) and (0.92 V, 0.77 V) for SPCE/GPE/ZnO NPs, SPCE/LPE/ZnO NPs, and SPCE/OPE/ZnO NPs, respectively.

## 4. Conclusions

In this work, three citrus species were used in the synthesis of ZnO NPs in the presence NaOH and were characterized as summarized in Table 4 and Table 5. Results obtained were very comparable to those obtained in other literature reports, further validating the successful synthesis of the nanoparticles. Subsequently, the ZnO NPs were examined for electrochemical redox response on SPCE. We reported in details for the first time to the best of our knowledge, the rudimentary electrochemical redox behavior of pure ZnO NPs derived from plant waste peel extract. The bare electrode showed enhanced performance following modification with ZnO NPs, and experienced an increase in electroactive surface area from 0.82 to 2.48, 2.20 and 1.86 mm^2^ for the SPE/GPE/ZnO NPs, SPE/LPE/ZnO NPs and SPE/OPE/ZnO NPs, respectively (Table 6). Hence, the economy of nanotechnology can benefit more from the eco-friendliness of ZnO NPs and the management of green waste.

## Figures and Tables

**Figure 1 materials-13-04241-f001:**
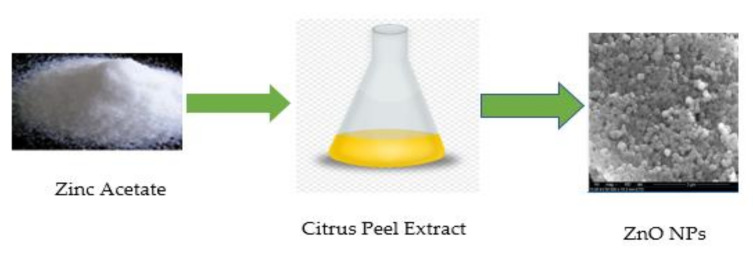
Schematic representation of ZnO NPs synthesis.

**Figure 2 materials-13-04241-f002:**
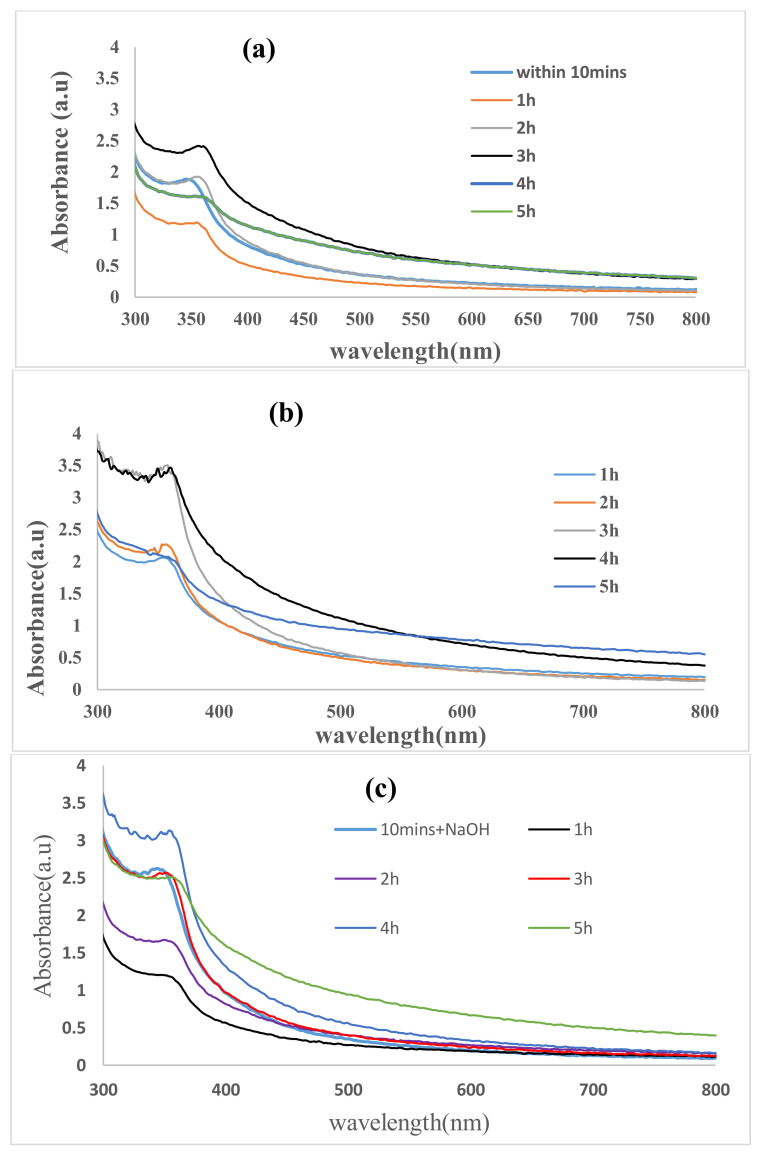
Absorption spectrum of ZnO nanoparticles produced from (**a**) Grape, (**b**) Lemon and (**c**) Orange peel extract within 10 min, within 10 min of introducing NaOH, 1 h, 2 h, 3 h and 4 h.

**Figure 3 materials-13-04241-f003:**
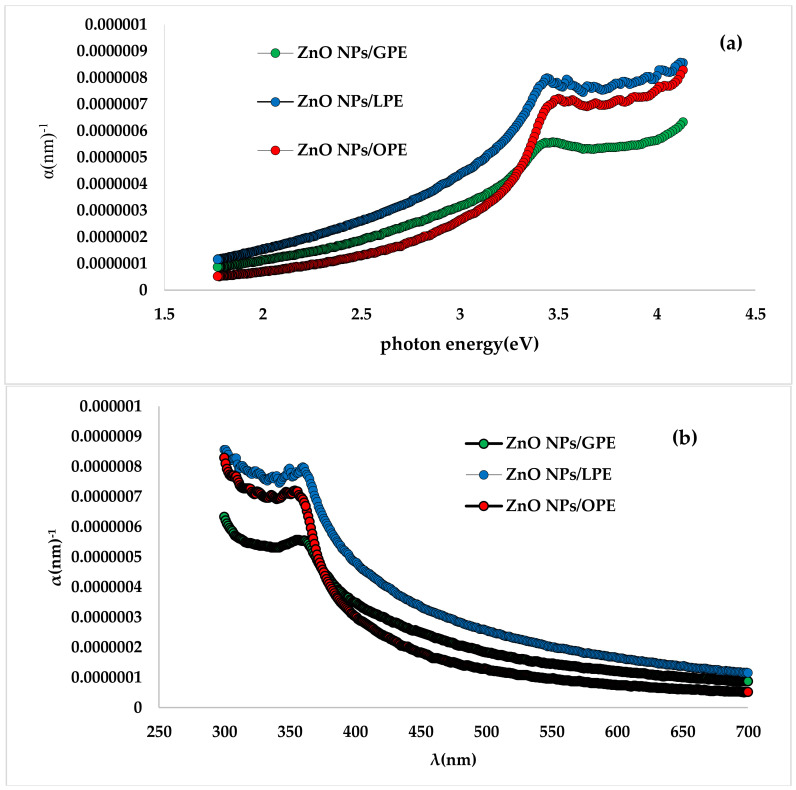
Dependence of the absorption edge on (**a**) photon energy (h) and (**b**) wavelength (λ).

**Figure 4 materials-13-04241-f004:**
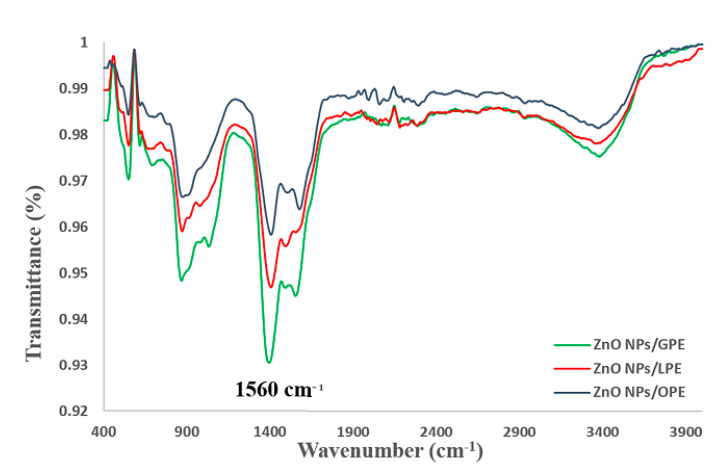
FTIR spectra of CPE/ZnO.

**Figure 5 materials-13-04241-f005:**
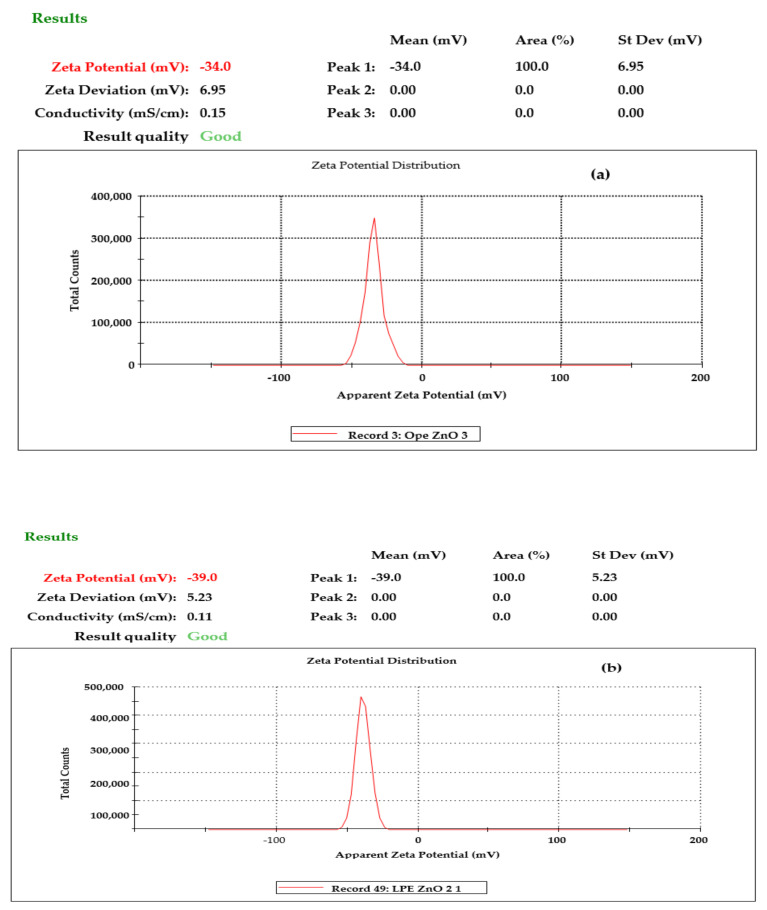
The ζ result of ZnO NPs from (**a**) OPE (**b**) LPE and (**c**) GPE (**d**) showed the comparative zeta potential and particle sizes of the ZnO NPs from the peel citrus extracts.

**Figure 6 materials-13-04241-f006:**
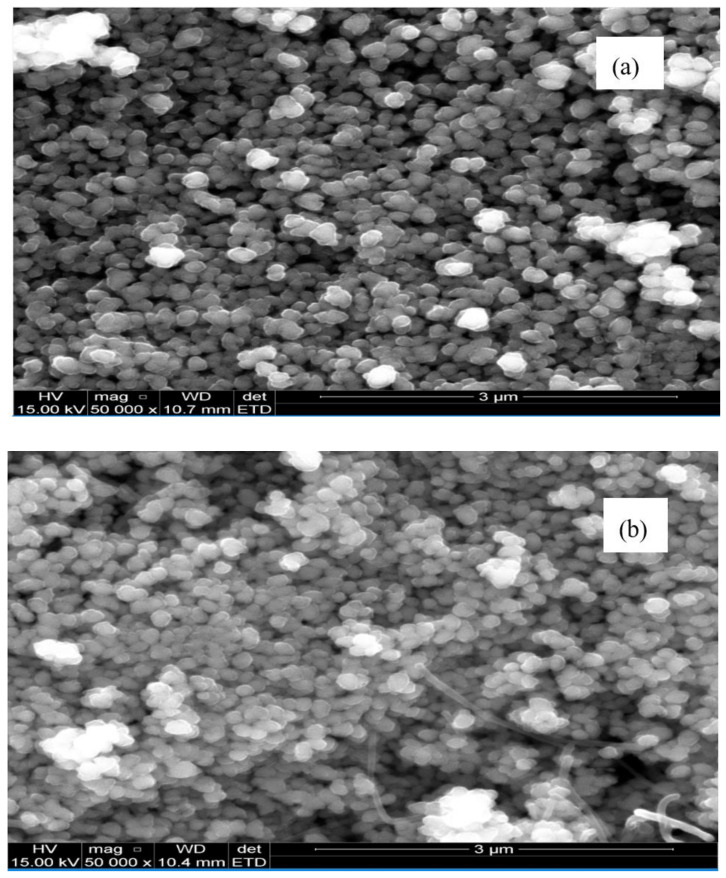
The SEM images of (**a**) GPE/ZnO NPs, (**b**) LPE/ZnO NPs and (**c**) OPE/ZnO NPs.

**Figure 7 materials-13-04241-f007:**
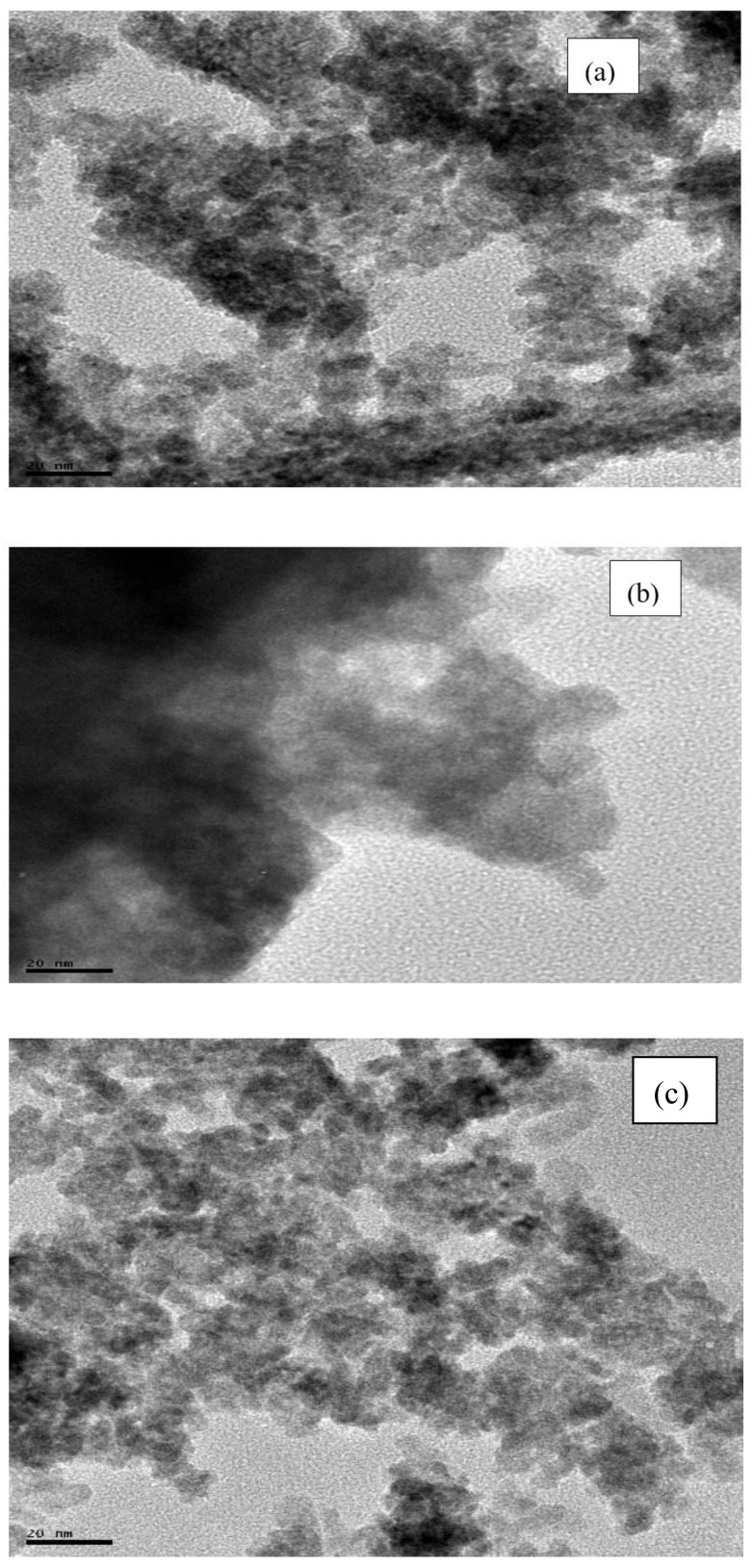
TEM images of (**a**) GPE/ZnO NPs, (**b**) LPE/ZnO NPs (**c**) OPE/ZnO NPs.

**Figure 8 materials-13-04241-f008:**
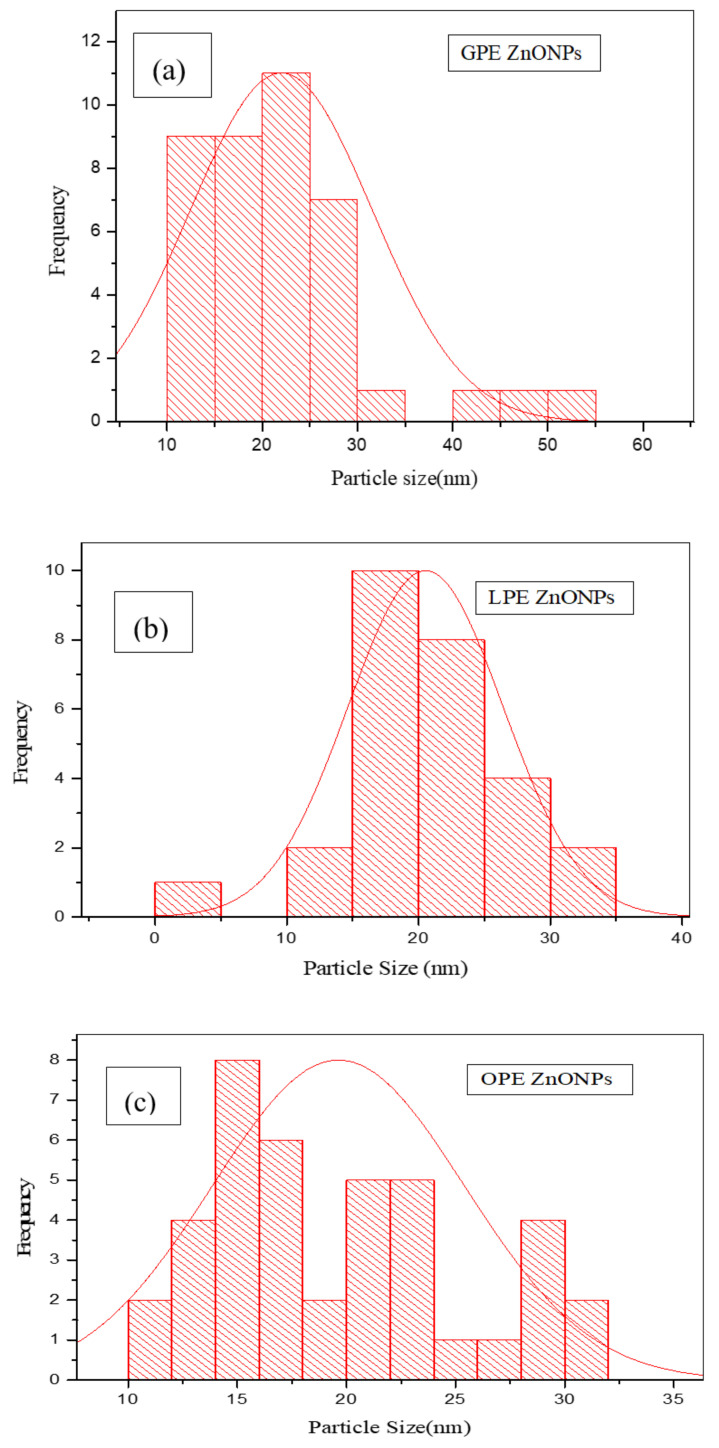
Size distributions of (**a**) GPE/ZnO NPs, (**b**) LPE/ZnO NPs (**c**) OPE/ ZnO NPs.

**Figure 9 materials-13-04241-f009:**
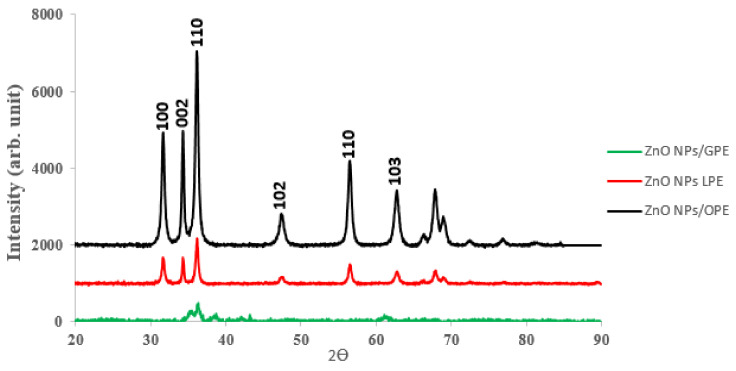
The XRD patterns of the ZnO NPs the three citrus peel extract.

**Figure 10 materials-13-04241-f010:**
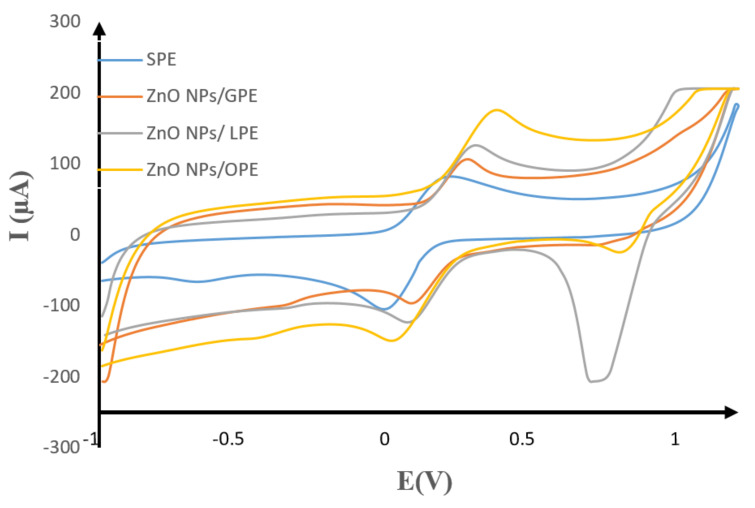
Comparative cyclic voltammograms of 10 mM K_3_[Fe(CN)_6_] in 0.1 M KCl at pH ≈ 7 on the bare SPE (blue) and modified SPCE/GPE/ZnO NPs (orange), SPE/LPE/ ZnO NPs (ash), and SPE/OPE/ ZnO NPs (yellow) at the scan rate of 100 mV/s.

**Figure 11 materials-13-04241-f011:**
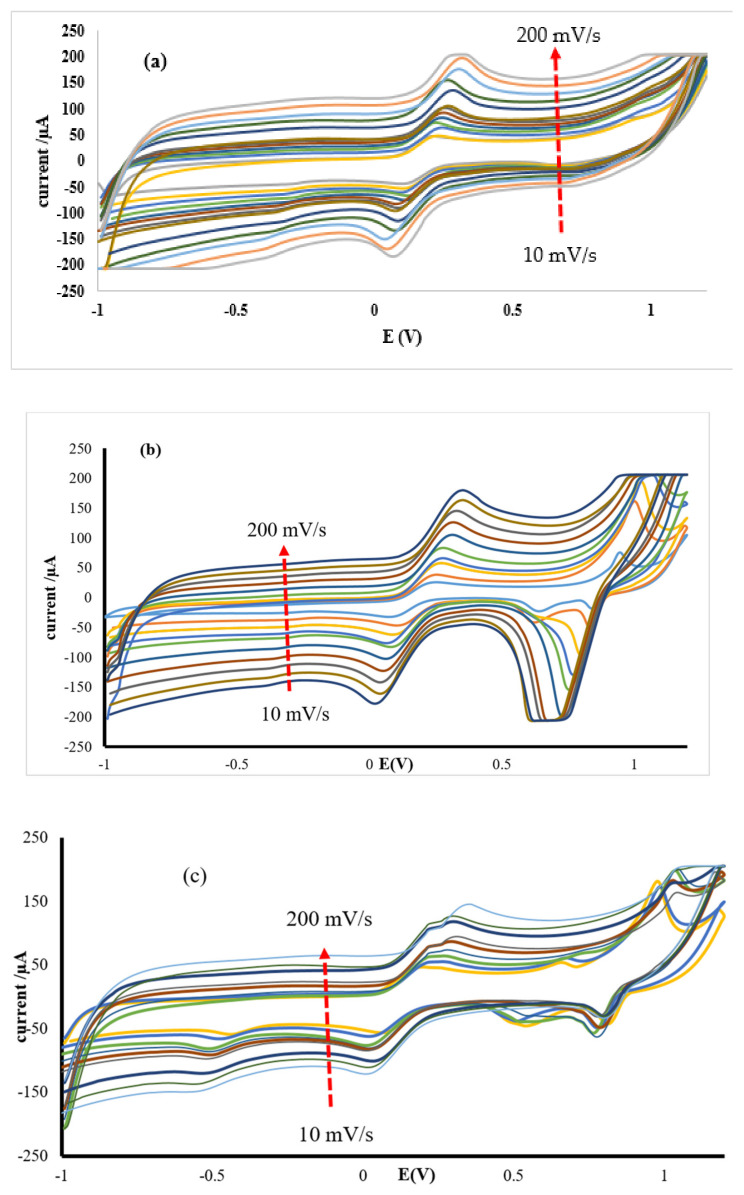
The effect of scan rate (10–200 mV/s) on cyclic voltammograms of 10 mM K_3_[Fe(CN)_6_] in 0.1 M KCl at pH ≈ 7 for (**a**) SPCE/GPE/ZnO NPs, (**b**) SPCE/LPE/ZnO NPs and (**c**) SPCE/OPE/ZnO NPs.

**Figure 12 materials-13-04241-f012:**
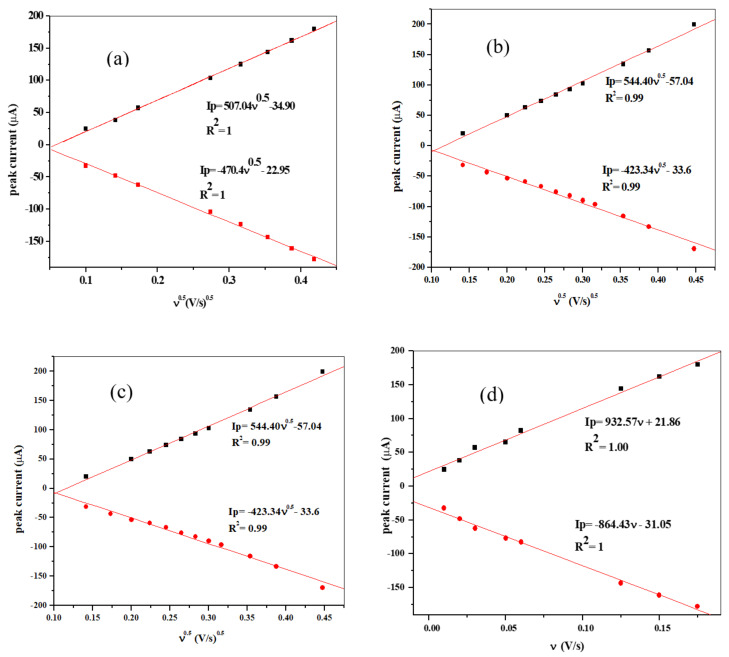
The plot of Ip (current) versus square root of scan rate (**a**–**c**) and Ip (current) versus of scan rate. (**d**–**f**) for the SPCE/LPE ZnO NPs, SPCE/GPE/ZnO NPs and SPCE/OPE/ZnO NPs respectively in 10 mM K_3_[Fe(CN)_6_] prepared in 0.1 M KCl at pH ≈ 7.

**Figure 13 materials-13-04241-f013:**
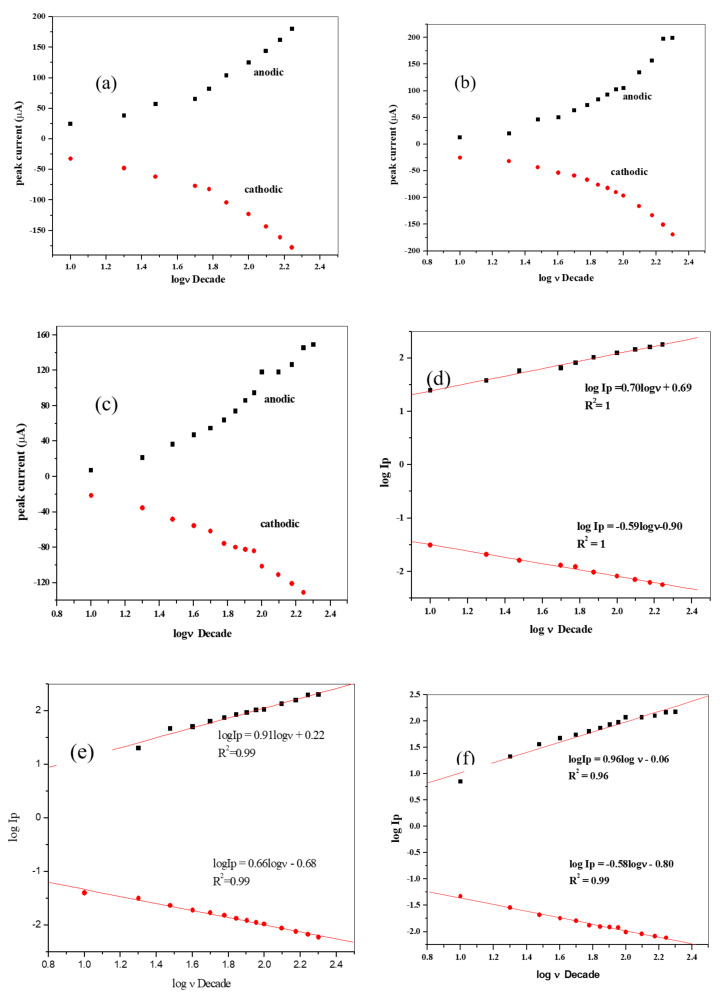
The plot of log of peak current vs. scan rate for (**a**) SPE/LPE/ZnO NPs, (**b**) SPE/GPE/ZnO NPs, and (**c**) SPE/OPE/ZnO NPs and log of peak current vs. log of scan rate for (**d**) SPE/LPE/ZnO NPs, (**e**) SPE/GPE/ZnO NPs, and (**f**), SPE/OPE/ZnO NPs in 10 mM K_3_[Fe(CN)_6_] in 0.1 M KCl at pH ≈ 7.

**Figure 14 materials-13-04241-f014:**
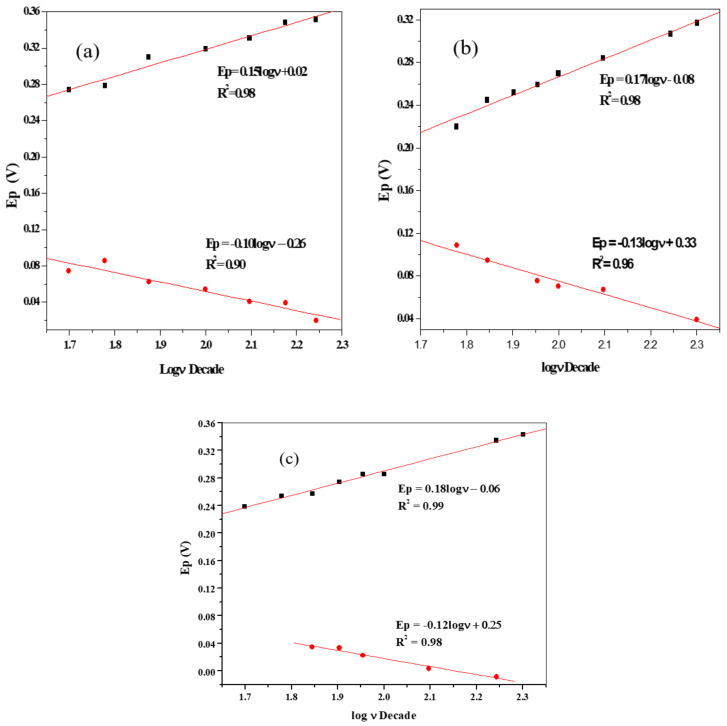
The plot of log of peak potentials vs. scan rate (**a**–**c**) for the SPCE/LPE/ZnO NPs, SPCE/GPE/ZnO NPs, and SPCE/OPE/ZnO NPs, respectively in 0.1 M KCl/10 mM K_3_[Fe(CN)_6_] at pH ≈ 7.

**Figure 15 materials-13-04241-f015:**
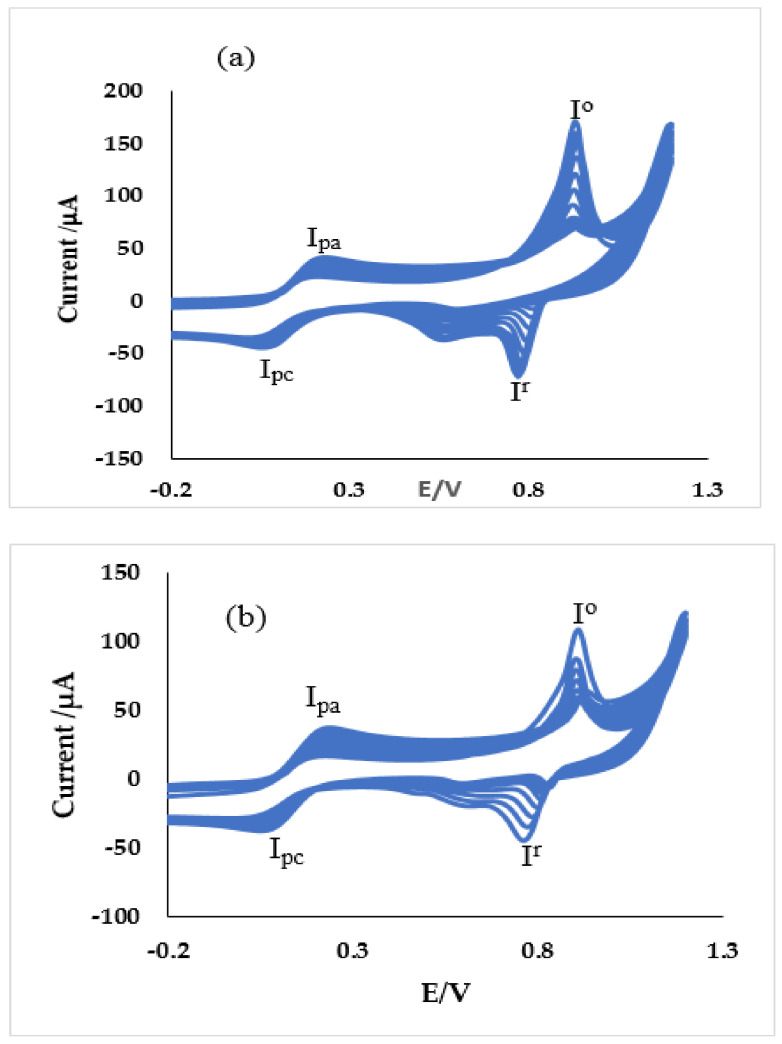
Current response (20 scans) of of (a) SPE/GPE/ZnO NPs/GPE, (**b**) SPE/LPE/ZnO NPs and (**c**) SPE/OPE/ZnO NPs in 10 mM K_3_[Fe(CN)_6_] prepared in 0.1 M KCl at pH ≈ 7 at the scan rate of 25 mV/s.

**Table 1 materials-13-04241-t001:** The Elemental Composition of the synthesized CPE/ZnO NPs.

Material.	Oxygen (%)	Zinc (%)	Total (%)
ZnO NPs/GPE	28.95	71.05	100
ZnO NPs/LPE	24.39	75.61	100
ZnO NPs/OPE	26.67	73.33	100

**Table 2 materials-13-04241-t002:** Spacing of the planes (dhkl) from XRD, JCPDS data card for equivalent (hkl) planes.

S/no	hkl	dXRD (A°)	dJCPDS(A°)	% of Contraction in d with JCPDS
1.	002	2.6048	2.6033	0.0576
2.	100	2.8202	2.8144	0.2057
3.	101	2.4803	2.4798	0.0202
4.	102	1.9134	1.9111	0.1202
5.	103	1.4787	1.4772	0.1014
6.	110	1.6283	1.6249	0.2088

**Table 3 materials-13-04241-t003:** The data of the redox peaks of 10 mM K_3_[Fe(CN)_6_] in 0.1 M KCl at pH ≈ 7 on the bare SPE, SPCE/GPE/ZnO NPs, SPCE/LPE/ZnO NPs and SPCE/OPE/ZnO NPs at the scan rate of 100 mV/s.

Electrode	I_pa_(µA)	E_pa_(V)	I_pc_(µA)	E_pc_ (V)	$E^{1/2}=\;\frac{E_{pa}+E_{pc}}{2}(V)$	$\frac{i_{pa}}{i_{pc}}$	∆E_p_(V)
SPE	85.17	0.19	−108.13	−0.02	0.09	0.79	0.21
SPE/ZnONPs/GPE	106.97	0.23	−94.98	0.06	0.15	1.13	0.17
SPE/ZnONPs/ LPE	125.03	0.32	−123.19	0.05	0.19	1.01	0.27
SPE/ZnONPs/OPE	175.08	0.38	−149.32	0.01	0.195	1.17	0.37

**Table 4 materials-13-04241-t004:** Comparative spectroscopic properties of ZnO NPs/CPE with other literature reports.

	UV-Vis (nm)	ζ (mV)	Eg (eV)	FTIR Peak cm^−1^	Technique	Reference
ZnO NPs/GPE	366	−42.9	3.26	553	Green	This work
ZnO NPs/LPE	360	−38.8	3.20	553	Green	This work
ZnO NPs/OPE	359	−34.2	3.30	553	Green	This work
ZnO NPs	-	-	-	450–540	Green	[13]
	-	-	-	540–417	Chemical	
ZnO NPs	215			-	Green	[1]
ZnO NPs	274	−51.8		~417	Green	[2]
ZnO NPs	-	-		668.29	Green	[5]
ZnO NPs thin film	363	-	--	470	Sol-gel	[8]
ZnO NPs	360	-	-	520	Sol-gel	[12]
Zinc nano rod	~370	-	3.37	-	Sol-gel	[36]

**Table 5 materials-13-04241-t005:** Comparative morphological study of ZnO NPs/CPE with other literature reports.

	Particle Size (nm)	EDX (% of Z/O Ratio)	d_101_ (2θ)	SEM	D(xrd) (nm)	Reference
ZnO NPs/GPE	~11–55	71.05:28.95	~36.25°	spherical	30.28	This work
ZnO NPs/LPE	~10–40	75.6:24.39	~36.25°	spherical	21.98	This work
ZnO NPs/OPE	~10–32	73.33; 26.67	~36.25°	spherical	18.49	This work
ZnO NPs	100–190	55.92:44.08	-	irregular	-	[13]
	100–200	68.30:31.70	-	nanoflowers	-	
ZnO NPs	9–10	-		net like	-	[1]
ZnO NPs	15–50	64.12:35.76	37°		31.8	[2]
ZnO NPs	11–25	-	36.19°	hexagonal	13.86	[5]
ZnO NPs thin film	10	-	36.18°	spherical	18.4	[8]
ZnO NPs	-	-	-	-	4	[12]
Zinc nano rod	68–116.	-	-	rodlike	1.6	[69]

**Table 6 materials-13-04241-t006:** Summary of electrochemical parameters of SPC/ZnO NPs/CPE electrodes.

	A (mm^2^)	n_α_	α	n_α_ α	ks, (s^−1^)	Γ (nmol cm^−2^)	(I_p_) vs. Log υ Slope	b (mVdec^−1^)
**ZnO NPs/GPE**	2.38	0.97	0.60	0.58	0.0059	2.037	0.7	300
**ZnO NPs/LPE**	2.20	0.79	0.57	0.45	0.0750	1.973	0.91	340
**ZnO NPs/OPE**	1.86	0.81	0.60	0.49	0.0021	1.578	0.96	360

## Data Availability

The data that support the findings of this study are available from the corresponding author upon reasonable request.

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
