# Peer review of "Green Wastes Mediated Zinc Oxide Nanoparticles: Synthesis, Characterization and Electrochemical Studies"

_materials, 2020, doi:10.3390/ma13194241_

Round 1
Reviewer 1 Report
The paper is to long, contains too many insignificant details. It should be substantially shortened. It is full of grammatical, typing and logical errors, bad figures and tables. It must be completely revised according to notes inserted as yellow notes into pdf of submitted paper.

Author Response
The authors would like to thank the reviewer for this comment. The response to the reviewer for the comment has been upload.

Reviewer 2 Report
- As the bandgap of ZnO is 3.37eV(L110), what cause the absorption peak at 359nm(L104) which with corresponded energy 3.45eV?
- In SAMPLE preparation, the procedure of controlled 10% v/v (L392) should be described clearly. Is there any cutting or drying procedure in the peels treatment?
- As ZnO NPs is formed by the reaction between zinc acetate and sodium hydroxide, is there any material data or references shows the properties for ZnO NPs prepared without these peels?
- The procedure of bare SPE(L240, 245) should be well explained. The setup for the CV measurement (L237) should be well described. What are the electrodes?
- Figure 6 is not clear. The procedure use TEM analysis to fine the average particle sizes(L181) and size distribution (L185) should be shown well in the text and/or in the figures.
- In extracting electron transfer coefficients (alpha)(L308), there is an inconsistency between the text (peak potential Ep(L307)) and the corresponded figures (Fig.12(a),(b),(c)), peak current ip (L312-L314)). The origin of eq.(15) should be referenced. The premises in using this eq. should be discussed.
- The parameters used in active surface area evaluation (L292) in eq.(14) should be shown apparently and discussed.
- With the ZnO NPs lattice constant evaluation(L234), what is the lattice constant differences for samples prepared from GPE, LPE and OPE? Why the XRD spectrum of ZnO NPs/GPE is difference to the others shown in Fig.8? What is the origin of these peaks? Does it take effect to the small Ipa and delta_Ep shown in Table 3(L258)?
- In Table 2, the authors show the lattice constant d calculated from different planes. With "% of contraction in d with JCPDS" shown in this table, certain discussion is necessary. Does it mean the lattice distortion? Besides, with this varied d, what is the effects to lattice constant a 1.9847A(L230)?
- With average size evaluated from TEM(L184), the LPE/ZnO sample shows the maxima average size. With mean size evaluated from XRD, the GPE/ZnO showS the maxima size. What is the origin of the difference? Please explain. Does this take effect on the properties described in this study?
- The “quasi-reversible one-electron process” with high delta_Ep character (L247) should be referenced.
- The origins of ZnO lattice constants (L232 and L232) should be referenced.
- What is the position of (102) and (103)(Table 2) in XRD spectrum (Fig 8)?
- Figure 15 is not clear.
- The corrections for eq.(5),(6),(7),(8) is necessary.
- What is Table 3040 in L197?
- The description format in L293-L298 is inconsistent with the formal format as in L215-L216.
- The figures of Fig.11(d),(e),(f) are same.
- What is “Dec” in L350-L352.
- What is “R” in eq.(13)? What is R^2 in L338?
- What is the origin of eq.(15)(L321)?
- The scan steps in Fig.10 should be labeled.
- What is the unit of v0.5 in Fig.11((1),(b),(c))?
Author Response

(The authors gave the same response as above.)

Reviewer 3 Report
A work presented for review entitled „Green wastes mediated zinc oxide nanoparticles: Synthesis, characterization and electrochemical studies” describes thriving green mediated reduction of zinc salt to ZnO NPs.
Overall, this manuscript is written clearly. The study design is clean and the results are convincing. I have just a few comments on this manuscript.
Line 112, 113, 214, 218-220, 290, 321, 336, 346: The formulas should be placed and described in the research methodology. There are no abbreviations in the formulas under the formulas.
Line 116-118: The graphs (a, b, c) presented in Figure 1 should have the same scales: absorbance 1-4, wavelength 0-800 and size.
Line 143, 144, 147, 198,410, 412: cm-1 change to cm-1
Line 160-163: Fig 4 - the results of 4a, 4b, 4c for OPE, LPE, GPE can be placed in one table
Line 234 and 257: Tab 2 and Tab3 - explain the abbreviations used under the table
Line 380: the chemical formula used should be corrected
To summarize: submitted paper can be accepted for publication after minor revisions.
Author Response

(The authors gave the same response as above.)

Round 2
Reviewer 1 Report
The paper can be published after minor revision reflecting comments inserted as yellow notes into pdf of submitted revised version of the paper.

Reviewer 2 Report
The manuscript is acceptable for publication.
Author Response
Thank you for your work and a positive assessment of our paper.